# Beneficial Influence of Exendin-4 on Specific Organs and Mechanisms Favourable for the Elderly with Concomitant Obstructive Lung Diseases

**DOI:** 10.3390/brainsci12081090

**Published:** 2022-08-17

**Authors:** Magdalena Figat, Grzegorz Kardas, Piotr Kuna, Michał G. Panek

**Affiliations:** Department of Internal Medicine, Asthma and Allergy, Medical University of Lodz, 90-419 Lodz, Poland

**Keywords:** Ex-4, neurodegeneration, Alzheimer’s, hypoxia, anti-inflammation, elderly, COPD, asthma

## Abstract

Exendin-4 (Ex-4), better known in its synthetic form and used clinically as exenatide, currently applied in the treatment of diabetes, induces a beneficial impact on nerve cells, and shows promising effects in obstructive lung diseases. At an advanced age, the development of the neurodegenerative process of brain tissue is masked by numerous concomitant diseases. The initial latent phase of neurodegenerative disease results in occurrence of manifestations at an advanced stage. To protect the brain and to simultaneously ensure proper treatment of common coexisting conditions in late life, such as diabetes, chronic obstructive pulmonary disease, or asthma, a pleiotropic medication should be chosen. Molecular mechanisms of Ex-4 exert neuroprotective effects or lead to secondary neurogenesis. Additionally, Ex-4 plays an important role in anti-inflammatory actions which are necessary both in the case of asthma and Parkinson’s disease. Specific receptors in the lungs also reduce the secretion of surfactants, which decreases the risk of exacerbation in chronic obstructive lung disease. In a great number of patients suffering from diabetes, asthma, or chronic lung disease, there is a great potential for both treatment of the main condition and protection against brain neurodegeneration.

## 1. Introduction

Extended life expectancy has resulted in an increased interest in age-specific conditions, such as Parkinson’s and Alzheimer’s disease. The growing interest in these diseases affecting the geriatric population results in numerous studies that search for therapies of neurodegenerative disorders, using the mechanisms of Exendin-4 (Ex-4).

The latest publications on the effects of Ex-4 therapy clearly indicate that this drug could be used in the treatment of neurodegenerative diseases [1]. Some of these studies are already at very advanced stages, including ongoing analyses on patients [1]. Among them, Ex-4 therapy for Parkinson’s disease is a dominant one. Patients with Parkinson’s disease also constitute a large group of potential beneficiaries of the drug. It is estimated that there are currently about 1 to 2 such patients per 1000 in the general population [2].

Patients with Alzheimer’s disease are another target group. Currently, 5–7 million new cases of Alzheimer’s disease are diagnosed annually [3]. Neurodegenerative diseases often pose a diagnostic challenge as it is difficult to clearly identify the disease which is directly responsible for dementia. Of all 36.5 million cases of dementia diagnosed worldwide, most are probably related to Alzheimer’s disease [3]. Among patients presenting any neurodegenerative condition, who could usufruct the Ex-4 treatment, are also these diagnosed with amyotrophic lateral sclerosis (ALS). In the general population [4,5], the number of patients affected by ALS is 1.7–2.3/100,000 per year.

Numerous studies, driven with various material (animal models, post-mortem preparations, cell cultures), lead to the same conclusion: not only diabetes [4], which has been so far treated with Ex-4, might benefit from therapy with Ex-4 insertion. 

## 2. General Characteristics of Ex-4

Exendin-4 (Ex-4) was isolated from the saliva of the venomous lizard, Gila monster (*Heloderma suspectum*) [6]. It is composed of 39 amino acids and differs from Exendin-3 (Ex-3) by the substitution of amino acids 2 and 3. Replacement of Ser^2^-Asp^3^ with Gly^2^-Glu^3^ causes differences in the bioactivity of these proteins and a significant reduction in the ability of Ex-4 to interact with receptors for proteins from the family of vasoactive intestinal proteins (VIP). This change does not affect the ability to bind to glucagon-like protein-1 receptors (GLP-1R) [7]. The sequence of this protein is 53% compatible with that of the endogenous glucagon-like protein-1 (GLP-1) [7], and secondary and tertiary structures as well as chain interactions determine its lipophilic properties [8] (Table 1).

Exendin-4 is a glucagon-like protein-1 analogue. In physiological conditions, GLP-1 is produced by L-cells in the intestines in response to food entering the digestive tract [10]. The described process is a fragment of the gut–pancreatic axis, whose mechanism of action is conditioned by incretins, i.e., GLP-1 and gastric inhibitory peptide (GIP). The incretin effect is responsible for 50–70% of postprandial secretion of insulin [11]. In patients with type 2 diabetes mellitus (T2DM), this effect is significantly weakened resulting from β-cell malfunction and treatment is required [11].

Ex-4 also improves insulin sensitivity, reduces glycated haemoglobin levels after 13 weeks of administration, and decreases body weight [12]. The dual mechanism of action, i.e., its effect on both the transcription of proinsulin genes and secretion of pancreatic insulin reserves, allows Ex-4 to be taken on a long-term as well as short-term basis [7]. Reduced glycaemic levels are observed after four hours [12]. Based on the documented mechanisms demonstrated by GLP-1 in in vitro conditions or in animals, the following conclusion was established also for in vivo conditions, which requires further detailed verification. In summary, Ex-4 would be ideal for many patients with T2DM and other civilization diseases, such as those associated with hypercholesterolemia [12].

Ex-4 can be administered in several forms. The most common method of application in the treatment of T2DM is by subcutaneous injection once a day or once a week. When administered intravenously, Ex-4 shows a stronger insulinotropic effect than GLP-1 [13] and crosses the blood–brain barrier in 90% of a given dose [8]. A higher percentage of the intravenous dose reaches the cerebral tissue than in the case of periventricular administration to the brain [8]. Such a high penetration of Ex-4 to the brain, given intravenously, allows the dose to be limited and avoids possible side effects. Injections exceeding the high dose of Ex-4, i.e., over 5 ug in a mouse, are likely to inhibit the entry to the brain. Considering these results, further research should be conducted to confirm or establish the inhibiting dose of Ex-4 in the human population as well [8]. Another possible way of application is the intraperitoneal method. Ex-4 administered intraperitoneally demonstrates prolonged hypoglycaemic activity [13]. Depending on the administration and the size of the dose, the intracellular accumulation of cyclic adenosine monophosphate (cAMP) inside the cell and hypoglycaemic activity can be modulated to some extent [14].

Passage via the blood–brain barrier is based on passive transport through simple diffusion because it does not cause energy consumption. It is not sensitive to manipulation by pharmacological or physiological factors either [8]. On the other hand, it is susceptible to physical and chemical factors such as lipophilicity and the ability to form hydrogen bonds. The low hydrogen bonding potential, characteristic of Ex-4, is an advantage while moving to a highly non-polar region of the cell membrane [8]. Ex-4, capable of crossing the blood–brain barrier, plays a very important role in the treatment of neurodegenerative diseases, and the spread of determined Ex-4 in the cerebral tissue after intravenous and periventricular administration is comparable [15]. This means that regardless of the method of administration, the drug will reach the same regions of the brain.

GLP-1 receptors, whose agonist is Ex-4, are located in many organs of the human body. Their presence has been confirmed in the pancreas, lungs, stomach, small intestine, kidneys, heart, most areas of the brain [16], and on peripheral nerves, i.e., the vagus nerve [17], and in the spinal cord [18,19,20] (Figure 1). They have not been found in the liver, skeletal muscles, or adipose tissue, i.e., in organs responsible for glucose metabolism in the body [21]. 

GLP-1R has also been discovered in the tracheal mucosa, which could create a chance for its potential application in the treatment of respiratory diseases. The Ex-4 ability to prevent bronchial hyperresponsiveness has been already evidenced on human isolated airway cells after high-glucose stimulation and sensitization through GLP-1R [22]. Additionally, these receptors have been traced in smooth muscles of the pulmonary artery responsible for functional pulmonary vasculature [7], enabling relaxation of its muscular layer [16,23,24]. In the pulmonary tissue, Ex-4 can contribute to a slight increase in mucus secretion through its insignificant ability to stimulate receptors for VIP [25] family proteins. Simultaneous action through different GLP-1 receptors and VIP family proteins is likely to ensure non-adrenergic and non-cholinergic peptide regulation of lung function.

Translation of the GLP-1R encoding gene results in the occurrence of 7-transmembrane-spanning, heterotrimeric G protein [26]. Initially, it is present on the cell surface. In response to GLP-1 or its analogues, it moves inside the cell and becomes activated through the membrane adenylate cyclase; however, it does not stimulate its release [6]. The activated adenylate cyclase, causing intracellular increase in cAMP, activates complex cascades of biochemical reactions and cAMP-dependent phosphokinase A (PKA), phosphatidylinositol kinase, and cAMP-response element-binding protein (CREB). The effects of the cascade depend on the type of specialization of the cell on whose surface the GLP-1 [27] receptor is located. 

The above-mentioned intracellular activation in pancreatic cells results in secretion of insulin from pancreatic reserves as well as deactivation of apoptosis-inducing enzymes. cAMP levels induced by Ex-4 are three times higher than after application of the same dose of GLP-1 [13]. Although the levels of both peptides return to baseline after 15 min, Ex-4 is more effective [13]. Ex-4 and GLP-1, if administered simultaneously, are additive to the intracellular cAMP concentration [7]. Due to a greater affinity of Ex-4 for the GLP-1 receptor than GLP-1, first the receptor binds to Ex-4 [7]. An increase in the cAMP level following Ex-4 administration is monophasic and starts at a dose of 100 pM. At a concentration of 10 nM, it turns into a *plateau*, and even above 100 nM, cAMP no longer grows, unlike in the case of Ex-3. This single peak is comparable to the first growth phase after the application of Ex-3 [6].

Ex-4 is a strongly binding ligand. After binding to GLP-1R, in a preparation made of the rat lung epithelium, Ex-4 did not unbind from the receptor even after the introduction of VIP family protein, histidine, or isoleucine [7] into the body. Ex-4 most easily binds to the receptor located in the lungs at pH = 7.4. In the lungs, the binding also depends on the concentration of the unbound ligand, i.e., the higher the concentration of the introduced substance, the better the binding to the receptor [28].

The effect of Ex-4 at the transcriptional level does not end with stimulation of proinsulin genes. The drug also intensifies the transcription of tyrosine hydrolysis in medullary catecholamine neurons. It is possible that changes in the concentration of tyrosine hydrolysis constitute a critical regulator of sympathetic influence involved in the regulation of physiological processes conditioning the cardiac function, including its arterial pressure and pulse values. An increased hydrolysis level leads to intensification of the sympathetic system activity in the regulation of physiological processes and to an increase in arterial pressure and heart rate. The observed changes in vital parameters are completely independent of glucose metabolism in the case of application of Ex-4 in T2DM therapy, and they may be considered significant side effects of this preparation [15]. Another aspect of the autonomic system activity, probably mediated through GLP-1 and its analogues, includes characteristic variability of the perception of interoceptive stress induced by, e.g., taste aversion, being a response to the detected taste of poison [15]. The occurrence of this reaction is largely evolutionarily protective and can be used in body weight reduction, where food will be given a specific taste and new visual and taste associations will be created. Potential effector pathways that regulate the sympathetic effect of incretinomimetics may be monosynaptic hypothalamo-spinal or bulbo-spinal, stimulated by sympathetic interstitial neurons [15].

Increased arterial pressure, accelerated heart rate, or taste aversion are not the only possible side effects of Ex-4. We should also be aware of potential gastrointestinal disturbances such as nausea and vomiting, resulting largely from delayed gastric emptying and pancreatitis [15,29]. Slackened peristaltic movements of the stomach and intestines, as well as longer retention of food, shorten time and degree of absorption of oral drugs [30]. This may lead to dangerous interactions between the administered preparations or intensification of adverse effects of the medications due to their long stay in the gastrointestinal tract (Table 2).

There are many clinical reasons for implementing Ex-4, e.g., as an incretinomimetic, which replaced GLP-1 in the treatment of diabetes. First of all, Ex-4 is highly resistant to dipeptidyl peptidase-IV (DPP IV), an enzyme responsible for degradation of GLP-1 and Ex-4 [15]. As a result, its half-life is about 120 min longer than that of GLP-1. The accompanying higher affinity for the GLP-1 receptor allows the maintenance of higher plasma insulin concentrations for a longer period of time [13]. Thus, maintaining the expected glucose blood concentration for a longer time without administering insulin injections several times a day allows the condition to be better managed by the patient [7,13]. The patients may also avoid mistakes resulting from improper adherence to the recommendations or treatment plan. Additionally, the drug is available in various forms for daily or weekly administration as a subcutaneous injection [31]. If the drug is administered in weekly doses, the patient does not experience hypoglycaemia even if the patient makes possible dietary errors as Ex-4 is not active in low glycaemia [7]. It induces postprandial secretion of insulin. After binding to the GLP-1 receptor, it stimulates the expression of proinsulin genes at the transcription level. The strongest effect on proinsulin promoter activity was recorded for 10 nM Ex-4 [7]. In the first phase of response to hyperglycaemia, i.e., the secretion of reserves from pancreatic cells, insulin reaches levels up to ten times higher than those observed in healthy individuals. Thus, insulin action is prolonged [13] and it can be administered in doses which are ten times lower [12]. The increase in insulin secretion is accompanied by an inhibited level of glucagon secreted by the liver, its sharp reduction, and a decrease in the level of free fatty acids [13].

An analysis of the most common application of Ex-4, the mechanisms of action used in T2DM therapy, receptor localization, factors favouring receptor binding, and distribution throughout the body suggest other potential applications of Ex-4 whose effects have been confirmed in studies on animal models (rodents: rats, gerbils) or in in vivo cell cultures.

## 3. Actions against Neurodegeneration of Cerebral Tissue

Currently, apart from its use in diabetes, many neuroscientists are working on the application of Ex-4 in the therapy of neurodegenerative diseases. This is evidenced by numerous studies on the use of Ex-4 in the treatment of Parkinson’s disease. 

In 2017, the Institute of Neurology and Neurosurgery in London published results of its randomized, double-blind trials conducted on patients with Parkinson’s disease [1]. The researchers observed that inclusion of Ex-4 into the therapy for a period of 48 weeks with a prior exclusion of medications previously used in Parkinson’s disease contributed to an improvement by 3.5 points on the third part of the Movement Disorder Society-Sponsored Revision of the Unified Parkinson’s Disease Rating Scale evaluating the patient’s motor symptoms. Additionally, the patients demonstrated smaller DaTScan binding loss in the putamen (bilaterally) and caudal nucleus on the right side, which was confirmed by imaging tests [1]. 

Theoretical assumptions described in the above-mentioned research, as well as in many other studies, are based on the hypothesis that Ex-4 prevents or inhibits the neurodegenerative process, which can be divided into several elements. The first of them is the neuroprotective effect. This is achieved by preventing the activation and expression of metalloproteinase-3 (MMP-3) in the substantia nigra of the mesencephalon and striatum. Any action or medication which stops or inhibits its expression promotes the reduction in dopaminergic neurodegeneration. This neuroprotective property more effectively prevents the activation of microglial cells in the mentioned structures of the encephalon [32]. Kim S. et al. also observed gene transrepression for TNF-α and IL-1β, i.e., several inflammatory mediators [32] which might be actively involved in pathogenesis of Parkinson’s disease. This fact is significant for prevention and treatment of neurodegenerative diseases, such as Parkinson’s disease, where an inflammatory response is highly important in the etiopathogenesis [33,34,35].

Moreover, an increase in the number of neuronal microtubules associated with protein 2, β-III-tubulin, and neuro-specific enolase with concomitant increase in the number of stem cells promotes neurogenesis even in adult individuals, which leads to further normalization of dopamine imbalance [36]. Significantly elevated levels of both tyrosine hydrolyses and the vesicular monoamine transporter 2 (VMAT2) in the substantia nigra were also noted [36]. The effect of Ex-4 was confirmed in animal model studies conducted in 2018 with the use of long-acting Ex-4 (PT302) in the form of biodegradable microspheres, administered subcutaneously once every two weeks. The beneficial effect of the drug is probably associated with the protection of the ends of tyrosine hydrolysis in the striatum and prolonged survival of dopamine neurons, even in the altered striatum [37].

In the course of Ex-4 therapy, an increase in the phosphorylated form of cofilin in nerve cells is observed. Phosphorylation of this protein, i.e., its deactivation, inhibits apoptosis induction, which results in the reduced mortality of these cells [26]. Moreover, the administration of Ex-4 before the introduction of cells into oxidative stress significantly reduces the number of cells which are subject to apoptosis [26]. Application of this protein increases the SH-SY5Y human neuroblastoma cell survival rate by 37% and, after administering two doses, toxicity of oxidative stress decreases by up to 73%. Consequently, it significantly reduces severity of the induction of programmed nerve cell death—apoptosis [38].

Ex-4 therapy appears to be beneficial for patients with neurodegenerative diseases, characterized by the degeneration of cholinergic neurons, such as amyotrophic lateral sclerosis (ALS). In this disease, the administration of Ex-4 reduces the loss of vitality and permeability of the cell membrane induced by H_2_O_2_ [5]. Along with a decreased growth of the activity of caspase-3, it protects the cell against apoptotic activity of oxidative stress. The most desirable action in this case is to increase the activity of choline acetyltransferase, which leads to promotion of the production of acetylcholine, being an important neurotransmitter in neural connections. This process is accompanied by a decrease in the level of lactate dehydrogenase (LDH), which is probably responsible for the protective effect or weakened damage to the motoneuron [5]. 

Secondary neurogenesis, which occurs after the completed development of nerve tissue in adults, is another important mechanism of action of Ex-4 which is involved in the inhibition or prevention of neurodegeneration. Ex-4 promotes the above process by increasing the number of microtubules associated with protein 2, β-III-tubulin, and neuro-specific enolase [36], as mentioned above. In addition, there is a significant increase in the proliferation and differentiation of cells in the subgranular layer of the dentate gyrus of the hippocampus [39]. This process was observed not only in healthy rodents on the Parkinson’s disease model, but also on the model of diabetes induced by a high-fat diet. The mechanism contributing to neurogenesis induced by such a diet has not been identified [31]. What is more, in other studies in which such a diet was used, the long-term potential (LTP) values were similar to those found in the age-matched control group. The results revealed an improvement in the cognitive function, which indicates an improvement in hippocampal cell plasticity [40]. These studies carried out on a human nerve cell showed that phosphatidylinositol kinase is involved in the neuronal differentiation induced by Ex-4 [26].

The third element is the ability of Ex-4 to reduce oxidative stress toxicity in nerve tissue. An animal model study (18-day Sprague-Dawley rat embryos) reconfirmed 73% mortality of primary hippocampal cells after using glutamate. Additionally, after administration of Ex-4, 25% of cells were affected by apoptosis. It follows that the other 75% were protected and stayed alive [40]. Furthermore, the above mechanisms revealed a decrease in H_2_O_2_ toxicity, inducing oxidative stress. This reduction may be achieved by deactivation of enzymes, i.e., phosphorylation of cofilin which induces apoptosis [26] or prolonged survival despite exposure to the same concentration of H_2_O_2_ [38].

The above-mentioned mechanisms justify the growing interest in Ex-4. Many studies deal with the implementation of the drug into therapy or prevention of neurodegenerative diseases. Various locations of the Ex-4 receptor encourage a search for further action profiles, apart from the one that can be used in neurology.

## 4. Protective Action of Ex-4 in Hypoxia of Various Tissues

Hypoxia resulting from impaired blood supply or lack of blood supply to the heart, brain, kidneys, or the pancreas affects tissue metabolism. Hypoxia can dysregulate gene expression, such as HIF-1α [41], or contribute to the degeneration of the involved tissue. As for nerve tissue, early administration of Ex-4 has a protective effect.

After the administration of Ex-4, hypoxic cells demonstrate substantially decreased activation of HIF-1α. Additionally, decreased expression of Bcl/Bax genes inhibits apoptotic pathways. Reduction in apoptosis following a prior Ex-4-induced reduction in HIF-1α expression indicates potential proapoptotic properties of this factor [42]. The effect of Ex-4 on HIF-1α expressions suggests that the expression of this factor is activated by a GLP-1R-dependent pathway [42].

Ischemia causing hypoxia and re-reperfusion of cerebral tissue also contributes to a change in cellular immunoreactivity. It starts to intensify after only six hours following the change in conditions and reaches its extreme value as soon as after one day. On the next day, it starts to decrease, and on the fourth day it becomes the least intensive. Its value stays the same for next few days. This leads to delayed neuronal death. It is only after ten days that the immunoreactivity of brain tissue cells gets back to normal. The exact mechanism of this phenomenon has not been explained. Application of Ex-4 before the episode of ischemia significantly reduces hyperactivity induced by the drug on the first day, which proves its protective effect on nerve tissue. Implementation of such a therapy also considerably reduces the activation of the microglia in the ischemic region [43]. 

The heart is another human organ which may suffer severe ischemia-related complications. A study conducted on cardiomyocytes confirms reports on the protective action of Ex-4 against damage caused by hypoxia. It does not only weaken hypoxia-induced apoptosis but also enhances intracellular expression of p38 mitogen-activated protein kinase (p38MAPK). This change is also accompanied by translocation of the glucose-4 transporter (GLUT-4). Ex-4 administration is associated with improved glucose uptake and increased production of ATP. Changes observed after application of Ex-4 include diminished cell destruction, which, consequently, results in cell viability, decreased LDH and creatine kinase (CK-MB) as well as reduced mortality. In addition, optimization of cell metabolism was noted [44].

Mechanisms inhibiting apoptotic processes play an important role in prevention of neurodegeneration and protection in hypoxic conditions. The mechanisms include, among others, glutathione peroxidase (GPxR) and thioredoxin reductase (TRxR). Their increased activity reduces generation of reactive oxygen species (ROS) even by 50%. Moreover, it prevents reduction in protein kinase b activity (AKT) and allows the maintenance of its level prior to the episode of ischemia [45].

Renal parenchyma tissue serves as a place where enhanced mRNA expression for anti-inflammatory factors, such as interleukin-10 (IL-10) and nitric oxide synthase (eNOS), occurs. Apart from various mechanisms of the GLP-1 receptor stimulation, a variable distribution of the receptor is observed. The different distribution depends on specific conditions. GLP-1R has an ability to autoregulate the expression after an acute ischemic episode. Moreover, there is a reverse correlation between the severity of kidney damage and expression of GLP-1 receptors in its parenchyma. This phenomenon stops deterioration of the renal function in the most severe cases. This is achieved by inhibition of macrophage recruitment, DNA damage, inflammation, and further production of ROS and other mechanisms inducing apoptosis, which intensifies the progress of kidney damage [46].

Having followed the publication of Waddell et al. [47] and Ahmad et al. [48], hypoxia results in exacerbation of asthma. Thus, Ex-4 insertion to the asthma treatment might decrease the worsening frequency and protect tissues affected by hypoxia.

## 5. Anti-Inflammatory Properties of Ex-4

Considering the definition and the underlying pathomechanisms of asthma, it appears that Ex-4 may be effective in the treatment of the disease. The mechanism described in the previous paragraph shows that Ex-4 weakens the expressions of TNF-α and IL-1β [32]. A 12-week therapy with Ex-4 administered at two doses of 10 µg in patients with T2DM included in a randomized and single-blinded study contributed up to 31% TNF-α suppression and 22% IL-1β suppression. Only a single application of Ex-4 brings significant effects for a patient. Four hours following the application of the drug, a decrease in the levels of TNF-α and IL-1β is observed (by 18% and 26%, respectively). It may be concluded that only a single dose can be administered, if the IL-1β suppression is to be achieved. Moreover, the drug inhibits ROS generation, nuclear transcription factor κB (NF-κB), suppressor of cytokine signalling 3 (SOCS-3), and other inflammatory markers, such as c-Jun N-terminal kinase (JNK-1), Toll-like receptor-2 (TLR-2), and Toll-like receptor-4 (TLR-4). All the above factors reveal a decrease in the expression, both after a long term (here, 6 and 12 weeks for ROS, NF-κB, TNF-α, and IL-1β, as well as 3 and 6 weeks for SOCS-3, JNK-1, TLR-2, TLR-4) and single administration. In the case of ROS, the effect is maintained after 12 weeks of therapy. Apart from inhibiting the expression of the above-mentioned factors, Ex-4 treatment simultaneously lowers the concentration of monocyte chemoattractant protein-1 (MCP-1), serum amyloid A (SAA), interleukin-6 (IL-6), and metalloproteinase-6 (MMP-6). No effect on IL-10 [49] was observed. With regards to other GLP-1R locations, obstructive pulmonary diseases could be potential targets of Ex-4 due to its presented mechanisms of action. 

Another reason for the use of Ex-4 in obstructive lung diseases is its ability to prevent bronchial hyperresponsiveness. In the course of various forms of asthma, patients often complain about triggering factors causing dyspnoea, which significantly reduces their daily functioning and decreases their quality of life. As shown in studies conducted on human bronchi under ex vivo conditions, GLP-1R stimulation leads to bronchial relaxation through the activation of cAMP protein kinase A (PKA) [22,50,51]. The exact mechanism of the obtained results has not been explained yet.

Combining the identified mechanisms of Ex-4 with those already used in the treatment of asthma would allow the intensification of the effects of anti-inflammatory drugs and reduce the severity of symptoms by decreasing bronchial hyperreactivity [22,51].

## 6. Beneficial Effects in Chronic Obstructive Pulmonary Disease (COPD)

Exposure to cigarette smoke or air pollution achieves a significant level not only in the big cities at present. Both contribute main risk factors to the development of chronic obstructive pulmonary disease (COPD) which is characterized by high mortality [44].

Research conducted on an innovative experimental animal model of lung diseases (a combination of a model of mice with asthma induced by exacerbation of the underlying disease through administration of ovalbumins and mice manifesting COPD during an exacerbation episode caused by lipopolysaccharides) undergoing Ex-4 therapy resulted in a significant reduction in the mortality of mice with COPD through the induction of bronchial relaxation, not only by relaxing smooth muscles through GLP-1 but also by myorelaxation of the muscle layer in the vascular bed [52]. It was contributed by a 3-fold increase in the mRNA expression for CD68 characteristics for macrophages and increased expression of MCP-1, IL-6, IL-10 [22,52]. 

In an in vitro experiment on type II pneumocytes taken post-mortem from donors prepared for transplantation, the mechanism by which surfactant secretion could be regulated [53] was investigated. Due to 53% homology of the Ex-4 structure with GLP-1(7–36) [7], the effects on the pneumocyte preparation were compared for each substance. The results appeared to be similar. Application of GLP-1(7–36) or Ex-4 stimulates secretion of phosphatidylcholine (the main component of surfactant), which is associated with the release of intracellular cAMP. The signal is transmitted via cAMP-dependent kinase A (PKA) and cAMP-dependent kinase C (PKC) [53], with no increase in intracellular Ca^2+^, which probably excludes the previously described mechanism of surfactant formation involving Ca-CM-PK [54]. Having plotted pleiotropic action of GLP-1 receptor agonists, i.e., affecting mucosal secretion, smooth airway muscles, and surfactant production, they act as hormones in the respiratory system [53]. In another study (2013) [52] analysing the effect of Ex-4 on surfactant secretion and conducted on the above described novel model of lung diseases, a significant improvement was noted in lung function of rodents affected by the induced exacerbation of lung disease. However, the improvement was not related to the expression of surfactant proteins or weakened expression of inflammatory markers. Four proteins (SFTP—surfactant proteins, i.e., SFTPA, SFTPB, SFTPC, SFTPD) were isolated from the surfactant. No changes in SFTPA expression were observed; SFTPB and SFTPC expression levels decreased three or four times after administration of Ex-4 and the SFTPD expression level increased, similarly to the one in the control group. None of the rodents treated with Ex-4 died [52]. Repeating the experiment on a preparation from human type 2 pneumocytes, isolated post-mortem with the methods used for determination of particular surfactant proteins, i.e., measuring methods, would allow a reliable comparison with results obtained from the human population.

Reduction in the expression of surfactant proteins, leading to a decrease in surfactant production and, as a consequence, to a decrease in the retention of surfactant proteins, may improve conditions for oxygen exchange in alveoli. It creates an opportunity to improve the normally already difficult oxygen shipment and to prevent the bacterial exacerbations in COPD. An enlarged vascular bed will be better supplied with blood which is responsible for oxygen transportation. Such a situation improves conditions of mechanisms reducing inflammation and oxaemia in the patient’s whole body. Reduced retention of sputum is also an important factor changing the conditions inside the alveoli in bacteriological terms. It is vital for prevention and treatment of bacterial infections in these patients, and often difficult to treat due to drug resistance.

## 7. Summary

The results of the publications presented above are the evidence of possible applications of Ex-4 in the treatment of age-related diseases (Table 3) (Figure 1 and Figure 2). Studies described in our review and concerning a potential application of Ex-4 in the treatment of neurodegenerative diseases are those most advanced ones. Results of numerous experiments carried out on animal models or cell colonies in vitro support their effectiveness in treatment of diseases such as Parkinson’s and Alzheimer’s disease or amyotrophic lateral sclerosis. Satisfactory results of the first randomized trials conducted on patients affected by these conditions promote the validity of these hypotheses.

The studies also focused on the influence of Ex-4 on inflammatory markers such as TNF-α, IL-1β, NF-κB, SOCS-3, JNK-1, TLR-2, and TLR-4. Experiments confirming the location of GLP-1 receptors in the lungs and clinical trials carried out on the above models of lung diseases open new therapeutic possibilities for another field of medicine which could benefit from the described mechanisms of Ex-4 action. Currently, Ex-4 preparations administered in patients due to diabetic indications are being observed for their beneficial effect on the lungs. The described results will turn out to be key information in further clinical trials, confirming the effectiveness of lung disease treatment with Ex-4.

Mechanisms triggered by Ex-4 active in hypoxia are also important. Depending on the time of its administration, the drug protects or minimizes damage caused by this traumatizing tissue condition.

In light of population aging and multimorbidity, in the near future this exogenous equivalent of GLP-1 may become a valuable and important drug due to its multidimensional mechanisms of action. Introduction of Ex-4 into therapy will allow the triggering of regenerative processes, especially in brain tissue, as well as neuroprotective processes which are important in old age due to the accompanying burden of neurodegenerative and cardiovascular diseases (e.g., atherosclerosis). The mechanisms responsible for gene transrepression of the above-mentioned inflammatory factors reduce the inflammation which is often chronic in people affected by many concomitant diseases. Additionally, the drug improves regeneration or reduces damage to organs affected by hypoxia. The variety of effects of Ex-4 may prove to be crucial in the treatment of patients with many coexisting diseases that are often challenging for clinicians.

## Figures and Tables

**Figure 1 brainsci-12-01090-f001:**
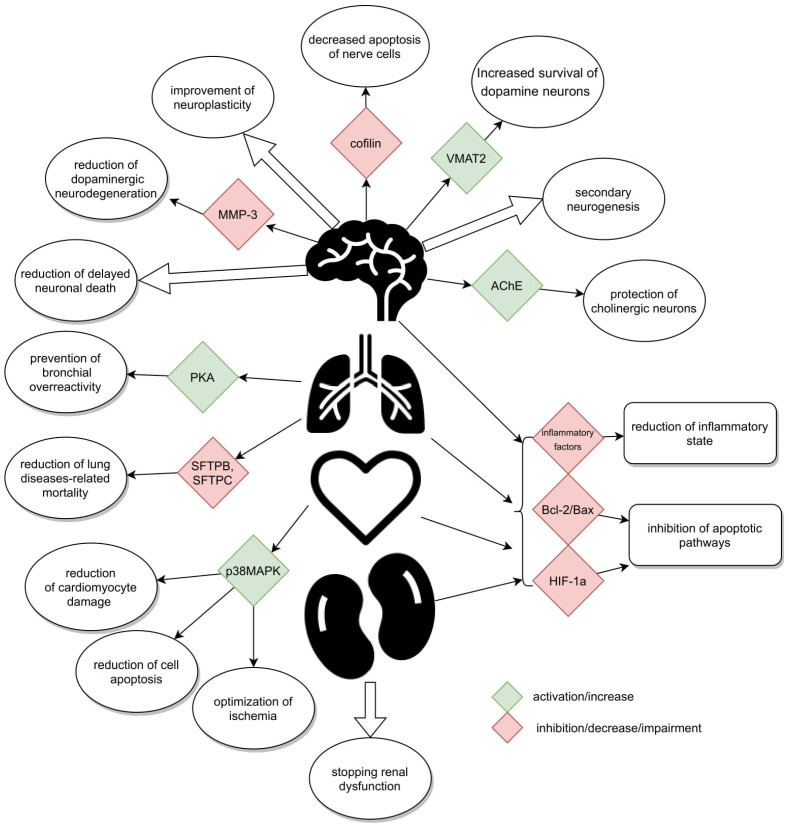
Schematically presented mechanisms.

**Figure 2 brainsci-12-01090-f002:**
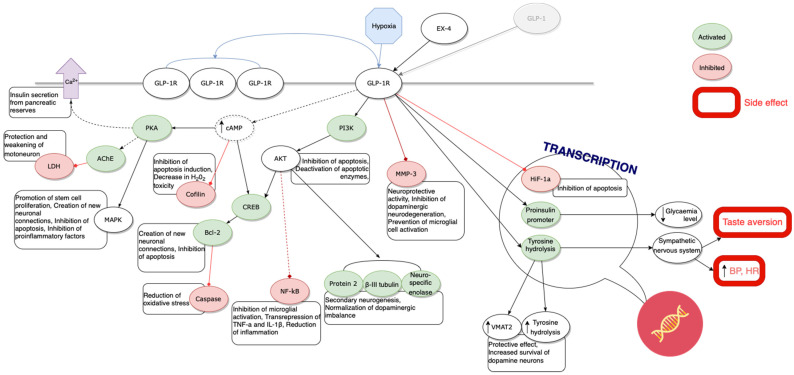
Summary of the presented mechanisms.

**Table 1 brainsci-12-01090-t001:** Comparison of GLP-1 and Ex-4.

Characteristics	GLP-1	Ex-4
Origin	Endogenous	Exogenous
Site of generation	L-cells in intestines	Salivary glands of *Heloderma suspectum* lizard
Structure	-	53% similar to GLP-1
Receptor	GLP-1	GLP-1, VIP proteins
Affinity to GLP-1R	-	Higher than GLP-1
Half-time	1.5–5 min [9]	120 min
Amount of cAMP secretion	-	Three times higher

**Table 2 brainsci-12-01090-t002:** Beneficial and adverse effects of Ex-4 in T2DM.

Beneficial Effects	Adverse Effects
Stimulation of proinsulin gene expression at the transcription level	Increased blood pressure
Release of insulin reserves from pancreatic cells	Increased heart rate
Inhibition of glucagon secretion by the liver	Aversion to taste
Improvement in insulin sensitivity	Delayed gastric emptying
Reduction of glycated haemoglobin levels	Slackening of peristaltic movements of the stomach and intestines
Body weight loss	Pancreatitis
No action at low glycaemic values	

**Table 3 brainsci-12-01090-t003:** Summary of the presented mechanisms.

Mechanism of Action	Potential Therapeutic Procedures
- Preventing the activation and expression of MMP-3, - Preventing the activation of microglial cells, - Occurrence of gene transrepression for TNF-α and IL-1β, - Promoting neurogenesis in adults, - Increasing tyrosine hydrolyses, increasing VMAT2, - Inhibiting apoptosis induction by phosphorylation of cofilin, - Reducing oxidative stress toxicity, - Increasing cell proliferation and differentiation, - Restoring initial LTP values	Prevention and treatment of neurodegenerative diseases
- Intensifying acetylcholine transferase activity, - Promoting acetylcholine production, - Reducing lactate dehydrogenase concentration	ALS prevention and treatment
- Decreasing the activation of HIF-1α factor, - Reducing Bcl/Bax gene expression	Treatment of complications due to organ ischemia
- Reducing ischemia-induced immunoreactivity increases in the first twenty-four hours, - Reducing microglial activation in the ischemic region	Treatment of ischemic strokes
- Enhancing p38MAPK intracellular expression, - Translocating GLUT-4, - Intensifying GPxR activity, reducing TRxR	Prevention of myocardial infarcts
- Self-regulating GLP-1R expression after acute ischemia, - Inhibiting macrophage recruitment	Prevention of complications of thromboembolism, vascular atherosclerosis
- Occurrence of gene transrepression for inflammatory factors (TNF-α, IL-1β, JNK-1, TLR-2, TLR-4, NF-κB, SOCS-3), - Activating PKA, preventing bronchial hyperactivity	Improvement in asthma control
- Relaxing bronchial smooth muscles, - Relaxing the muscle layer in the vascular bed, - Increasing mRNA expression for CD68 macrophages - Intensifying MCP-1, IL-6, IL-10 expression, - Decreasing SFTPB and SFTPC expression, increasing SFTPD expression	COPD treatment

## Data Availability

Data sharing not applicable.

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
