# Peer review of "Beneficial Influence of Exendin-4 on Specific Organs and Mechanisms Favourable for the Elderly with Concomitant Obstructive Lung Diseases"

_brainsci, 2022, doi:10.3390/brainsci12081090_

Round 1

Reviewer 1 Report

The article provides a balanced view of the potential clinical applications for exendin-4 (Ex-4), a glucagon-like protein 1 analogue currently used in the treatment of diabetes. Given the pleiotropic activity of the drug, treatment with Ex-4 may represent a valid therapeutic approach for coexisting pathologies- including neurodegenerative conditions- in the elderly. The relevant results in the field are described in a logical and clear manner, making the article easy to read; the figures and tables neatly summarize the presented mechanisms of action and/ or the therapeutic potential of the compound. The cited references were relevant.

Author Response

Dear Reviewer 1,
as an author team we are grateful to you for your review. Having noticed enormity work which was performed during the preparation of our work, the best kind of commendation was awarded to us. Hopefully the merits-related content will be appreciated also in next steps of submitting process resulting in publication presently. Thank you for sharing your time and decision.
Kind Regards,
Corresponding Author

Reviewer 2 Report

This manuscript reviews the beneficial effects of GLP-1 receptor agonism on neurological and pulmonary disease. This review is very well researched and mechanisms are thoroughly discussed in suitable depth. The Authors are to be commended for their efforts. However, I feel there is significant room for improvement regarding clarity, punctuation and clarity. At times I found it quite difficult to read as I was unsure what the Authors were trying to convey. My comments are as follows

Abstract

-       Please clarify that exenatide is the synthetic form of exendin-4 which is used clinically

-       The first two lines of the abstract do not flow well and these points could just be combined into a single point

-       Line 14: comma after ‘(Ex-4)’ and before ‘induces’

-       Line 15: proven, not proved. Perhaps remove the word proven/proved all together

-       Line 16: At an advanced age

1. Introduction

- Line 35-36: Perhaps Authors could provide a reference for readers. Remove ‘mainly’

- Lines 42-44: This is quite confusing. If 7 million new cases of AD are diagnosed annually, would there be more than 13.8 million people affected by 2050?

- Lines 48-49: This sentence does not make sense. Two things are being discussed at once

- Lines 50-53: These sentences do not flow well from previous point. Recommend providing an overview/summary sentence of conditions to be discussed in the review instead.

2. General Characteristics of Ex-4

- Lines 61-62: Use word ‘endogenous’ before GLP-1

- Table 1: All lines here should be capitalised. Is suspectumvenom a typo? It is referred to as suspectum in line 56. Be consistent with either arrows or words (i.e. amount of cAMP secretion should be described in words as other rows).

- Line 69: Provide source for statement

- Line 70: Why is effect significantly weakened in T2DM. Please provide explanation

- Line 73: ...administration, and decreases body weight

- Line 80: Reference for statement regarding hypercholesterolaemia

- Line 86: Reference required after ‘...administration to the brain.’

- Line 88: Is 5ug/mouse a correct unit? Perhaps say 5ug in a mouse

-  Lines 106-107: GLP-1 receptors also exist on peripheral nerves (many references you can select from) the vagus nerve (PMID: 34185884) and in the spinal cord (PMID: 24719110, 29329976, 34512747)

- Lines 110-111: The authors should elaborate on how mucosal receptors in the trachea could be beneficial? Do they mean bronchial receptors?

- Lines 119: Membrane-spanning

- Lines 128: It is a bit unnecessarily to bring up point regarding new neuronal connections in this sentence.

- Lines 139-140: Can this unbinding be harmful in the body?

- Lines 163-164: Pancreatitis is also a risk factor

- Table 2: All rows to be capitalised. I would not call inhibition of peristaltic movements of stomach and intestines an adverse effect – that is how satiety is achieved which aids with weight loss (one of the aims of GLP-1 treatment)

- Lines 178: Perhaps ‘psychologically accept the disease’ could be replaced by something more suitable (i.e. allows condition to be better managed by the patient)

- Lines 183: ‘he/she’ should be replaced by ‘the patient’

3. Actions Against Neurodegeneration of Cerebral Tissue

- Lines 214: What is black matter? Do you mean grey matter?

- Lines 218-220: Are TNF-alpha and IL-1beta involved in pathogenesis of PD? If so, include this point

- Line 237: Which human cells are you referring to?

- Line 239: Do you mean apoptosis?

- Line 242: Citation required for this statement.

- Line 268: Does that mean only 25% were affected by apoptosis and 75% were protected?

4. Protective action of Ex-4 in Hypoxia of Various Tissues

- Lines 281-284: citations required or do these all pertain to ref. 36?

- Lines 320: Unsure what you mean by ‘in the most severe damage.’

5. Anti-inflammatory Properties of Ex-4

- Lines 328-331: What were these effects? Did they have any clinical benefit (e.g. symptom improvement)?

- I feel like this paragraph and the next paragraph could be combined – they both discussion anti-inflammatory effects/benefits in obstructive pulmonary disease

6. Beneficial Effects in Chronic Obstructive Pulmonary Disease (COPD)

- Lines 357-359: This sentence requires more rewording. It is difficult to read

- Line 373: What is the ‘(7-36)’ in reference to? Are these sources? If so, they should follow the format of other sources in the paper [7-36]

- Lines 378-379: Perhaps better to say that they act as hormones in the respiratory system

- Line 380: What is the (2018) citation? Nothing else is given

- Lines 384: The connection between surfactants and COPD needs to be explained. It is brought up in lines 394-396 – bring these sentences forward

- Line 385: SFTP was not defined above. It is in the list of abbreviations though

7. Summary

- Line 407: Use the word ‘support’ instead of ‘prove’

- Line 410: Use the word ‘support’ (or similar) instead of ‘confirm’

- Table 3: Capitalise all rows

- Figure 1: Great figure. I feel like it will be useful to make reference to this earlier in the manuscript, if possible.

- Is Figure 2 referenced to in-text? I could not see it.

- Line 437: that are often challenging for clinicians. Remove comma after ‘diseases’.

Author Response

Dear Reviewer 2,

as a corresponding author on behalf of whole author team I am grateful to you for your words. I am thankful for your time and comments. Having noticed enormity work which was performed during the preparation of our work, the best kind of commendation was awarded to us. I did my best to improve pointed out places of the manuscript or to answer some of them. Please take a look and read the manuscript again. Hopefully it satisfies the publications conditions.

  1. Introduction

- Lines 42-44: This is quite confusing. If 7 million new cases of AD are diagnosed annually, would there be more than 13.8 million people affected by 2050?
-> The estimation in 2050 concerns all cases of neurodegenerative diseases, not only AD. But annually 7 million new cases are diagnosed with AD.

  1. General Characteristics of Ex-4

- Lines 139-140: Can this unbinding be harmful in the body -> In citied article is no date of harmfulness bound with this chemical feature. It is rather clearly chemical comparison.

- Lines 163-164: Pancreatitis is also a risk factor -> In this part possible sides effects of Ex-4 usage were discussed. In considered literature pancreatitis was not mentioned. I guess your suggestion was that pancreatitis, as a known condition by a patient, could be a risk factor to develop any adverse effects. Am I right?

  1. Actions Against Neurodegeneration of Cerebral Tissue

- Line 239: Do you mean apoptosis? -> yes

- Line 268: Does that mean only 25% were affected by apoptosis and 75% were protected? -> yes

  1. Protective action of Ex-4 in Hypoxia of Various Tissues

- Lines 320: Unsure what you mean by ‘in the most severe damage.’ -> I mean: more severe damage gives decreased expression of GLP-1R which leads to stopping the deterioration of kidney failure after ischemic trauma before the damage achieved the severe stage. The numerous GLP-1R activated by Ex-4 protects the renal tissue in the case of ischemic episode.

  1. Anti-inflammatory Properties of Ex-4

- Lines 328-331: What were these effects? Did they have any clinical benefit (e.g. symptom improvement)? -> yes, the effect is: Ex-4 provided significant protection for the kidneys against acute ischemia reperfusion injury.

- I feel like this paragraph and the next paragraph could be combined- they both discussion anti-inflammatory effects/benefits in obstructive pulmonary disease. -> Not exactly. The combination of those two paragraphs could obliterate effects which find implementation in COPD. These descripted in the next one. What’s more mentioned inflammatory factors are not only related to the COPD. They play also an important role in the pathomechanism of other diseases.

  1. Beneficial Effects in Chronic Obstructive Pulmonary Disease (COPD)

- Line 373: What is the ‘(7-36)’ in reference to? Are these sources? If so, they should follow the format of other sources in the paper [7-36]-> (7-36) is not a reference. It is a specification of GLP-1 used in the linked source.

- Line 380: What is the (2018) citation? Nothing else is given -> It is the year when the study was conducted. Unfortunately, any misspelling appeared there. It was 2013. reference is added.

  1. Summary

-„Figure 1: Great figure.” Nice to hear these words. Following your suggestion, the reference to this figure is put also in the 2. part, line 107.

The answers to other of your comments are inserted directly to the revised manuscript.

Thank you for your time and review.
Kind regards,
Corresponding Author

Round 2

Reviewer 2 Report

Please see file attached.

Author Response

Dear Reviewer 2,

as a corresponding author on behalf of whole author team I am grateful to you for your words. I am thankful for your time and comments. Having noticed enormity work which was performed during the preparation of our work, the best kind of commendation was awarded to us. I did my best to improve pointed out places of the manuscript or to answer some of them. Please take a look and read the manuscript again. Hopefully it satisfies the publications conditions.

Thank you for making these corrections/changes. Please see my responses below for each comment and if there is additional action required. Please note, there is still need for improvement in regard to grammar and punctuation throughout the manuscript. Perhaps the journal editors/publishers can assist with this. In future publications, when responding to reviewers, please address each comment individually with a response and where the change was made (and not just selected comments as seen below). This will aid the reviewer in localising all corrections and confirming they were all addressed.

Thank you again for your advices. It is really useful to know the preferred way of working form reviewer. I decided to remove the comment according the simple corrections to make the whole conversation easier to follow. This time I have it done as you suggested. Thank you for your time and review.

  1. Introduction

- Lines 42-44: This is quite confusing. If 7 million new cases of AD are diagnosed annually, would there be more than 13.8 million people affected by 2050? -> The estimation in 2050 concerns all cases of neurodegenerative diseases, not only AD. But annually 7 million new cases are diagnosed with AD.
Response: The sentence is still referring to AD as it says “affected by this neurodegenerative disorder”. Still unclear and please correct.
Author: To solve the situation I suggest to remove the one sentence. Correction has been done. Hopefully it is not confusing anymore.

  1. General Characteristics of Ex-4

- Lines 139-140: Can this unbinding be harmful in the body -> In citied article is no date of harmfulness bound with this chemical feature. It is rather clearly chemical comparison.
Response: Okay

- Lines 163-164: Pancreatitis is also a risk factor -> In this part possible sides effects of Ex-4 usage were discussed. In considered literature pancreatitis was not mentioned. I guess your suggestion was that pancreatitis, as a known condition by a patient, could be a risk factor to develop any adverse effects. Am I right?
Response: Sorry I was not clear. Pancreatitis is a (rare) side-effect of exenatide (https://pubmed.ncbi.nlm.nih.gov/19703814/) 2
Author: Additional adverse effect has been added in the text and in the table. New citation also. Thank you for your remark.

  1. Actions Against Neurodegeneration of Cerebral Tissue

- Line 239: Do you mean apoptosis? -> yes
Response: Okay but I recommend using this word.
Author: Correction is done.

- Line 268: Does that mean only 25% were affected by apoptosis and 75% were protected? -> yes
Response: Okay but please make this clear then
Author: Correction is done. I hope it is clear right now.

  1. Protective action of Ex-4 in Hypoxia of Various Tissues

- Lines 320: Unsure what you mean by ‘in the most severe damage.’ -> I mean: more severe damage gives decreased expression of GLP-1R which leads to stopping the deterioration of kidney failure after ischemic trauma before the damage achieved the severe stage. The numerous GLP-1R activated by Ex-4 protects the renal tissue in the case of ischemic episode.
Response: Okay but better to say “in the most severe cases”.
Author: Correction is done.

  1. Anti-inflammatory Properties of Ex-4

- Lines 328-331: What were these effects? Did they have any clinical benefit (e.g. symptom improvement)? -> yes, the effect is: Ex-4 provided significant protection for the kidneys against acute ischemia reperfusion injury.
Response: I was referring to improvement in clinical asthma (e.g. shortness of breath, cough, etc.). Please include if there was symptomatic benefit or not for asthma.
Author: A few sentences are added as a summary of this part. I hope it is the required reference to the asthma clinical performance. At present it’s the line 330-332.

- I feel like this paragraph and the next paragraph could be combined- they both discussion anti-inflammatory effects/benefits in obstructive pulmonary disease. -> Not exactly. The combination of those two paragraphs could obliterate effects which find implementation in COPD. These descripted in the next one. What’s more mentioned inflammatory factors are not only related to the COPD. They play also an important role in the pathomechanism of other diseases.
Response: Okay that is fair point.

  1. Beneficial Effects in Chronic Obstructive Pulmonary Disease (COPD)

- Line 373: What is the ‘(7-36)’ in reference to? Are these sources? If so, they should follow the format of other sources in the paper [7-36]-> (7-36) is not a reference. It is a specification of GLP-1 used in the linked source.
Response: Okay

- Line 380: What is the (2018) citation? Nothing else is given -> It is the year when the study was conducted. Unfortunately, any misspelling appeared there. It was 2013. reference is added.
Response: Thanks for correcting

  1. Summary

-„Figure 1: Great figure.” Nice to hear these words. Following your suggestion, the reference to this figure is put also in the 2. part, line 107.
The answers to other of your comments are inserted directly to the revised manuscript.
Response: Thanks

Thank you for your time and review.
Kind regards,
Corresponding Author